# Multi-Cohort Transcriptomic Subtyping of B-Cell Acute Lymphoblastic Leukemia

**DOI:** 10.3390/ijms23094574

**Published:** 2022-04-20

**Authors:** Ville-Petteri Mäkinen, Jacqueline Rehn, James Breen, David Yeung, Deborah L. White

**Affiliations:** 1Computational and Systems Biology Program, Precision Medicine Theme, South Australian Health and Medical Research Institute, Adelaide, SA 5000, Australia; 2Australian Centre for Precision Health, UniSA Clinical & Health Sciences, University of South Australia, Adelaide, SA 5000, Australia; 3Computational Medicine, Faculty of Medicine, University of Oulu, FI-90014 Oulu, Finland; 4Center for Life Course Health Research, Faculty of Medicine, University of Oulu, FI-90014 Oulu, Finland; 5Blood Cancer Program, Precision Medicine Theme, South Australian Health and Medical Research Institute, Adelaide, SA 5000, Australia; jacqueline.rehn@sahmri.com (J.R.); david.yeung@adelaide.edu.au (D.Y.); deborah.white@sahmri.com (D.L.W.); 6Faculty of Health and Medical Sciences, University of Adelaide, Adelaide, SA 5005, Australia; jimmymbreen@gmail.com; 7South Australian Genomics Centre, South Australian Health and Medical Research Institute, Adelaide, SA 5000, Australia; 8Robinson Research Institute, University of Adelaide, Adelaide, SA 5005, Australia; 9Australian and New Zealand Children’s Oncology Group, Clayton, VIC 3168, Australia; 10Department of Haematology, Royal Adelaide Hospital and SA Pathology, Adelaide, SA 5000, Australia; 11Faculty of Sciences, University of Adelaide, Adelaide, SA 5005, Australia; 12Australian Genomics Health Alliance, Parkville, VIC 3052, Australia

**Keywords:** acute lymphoblastic leukemia, RNA-seq, confounder adjustment, machine learning

## Abstract

RNA sequencing provides a snapshot of the functional consequences of genomic lesions that drive acute lymphoblastic leukemia (ALL). The aims of this study were to elucidate diagnostic associations (via machine learning) between mRNA-seq profiles, independently verify ALL lesions and develop easy-to-interpret transcriptome-wide biomarkers for ALL subtyping in the clinical setting. A training dataset of 1279 ALL patients from six North American cohorts was used for developing machine learning models. Results were validated in 767 patients from Australia with a quality control dataset across 31 tissues from 1160 non-ALL donors. A novel batch correction method was introduced and applied to adjust for cohort differences. Out of 18,503 genes with usable expression, 11,830 (64%) were confounded by cohort effects and excluded. Six ALL subtypes (ETV6::RUNX1, KMT2A, DUX4, PAX5 P80R, TCF3::PBX1, ZNF384) that covered 32% of patients were robustly detected by mRNA-seq (positive predictive value ≥ 87%). Five other frequent subtypes (CRLF2, hypodiploid, hyperdiploid, PAX5 alterations and Ph-positive) were distinguishable in 40% of patients at lower accuracy (52% ≤ positive predictive value ≤ 73%). Based on these findings, we introduce the Allspice R package to predict ALL subtypes and driver genes from unadjusted mRNA-seq read counts as encountered in real-world settings. Two examples of Allspice applied to previously unseen ALL patient samples with atypical lesions are included.

## 1. Introduction

Acute lymphoblastic leukemia (ALL) is characterized by abnormal differentiation and proliferation of malignant lymphoid precursors in blood and bone marrow [1,2]. Usually, the disease manifests as abnormal proliferation of B-cells while less than a quarter of patients present with a T-cell malignancy. The incidence rate is the highest in children under the age of 10 and in adults over the age of 65, with an average age-adjusted global annual incidence of 0.85 per 100,000 individuals [3]. Despite recent progress in molecular phenotyping, ALL remains a life-threatening disease, and advanced age (beyond pediatric) and certain subtypes are predictive of poor outcomes [1,2,4,5,6,7]. For these reasons, there is a compelling rationale for improving diagnostic tools and for pursuing deeper biological insight into ALL through new emerging technologies.

Cytogenetic testing, immunophenotyping and molecular assays are essential for the diagnosis and further stratification of the disease into subtypes with different biological characteristics and prognoses [8,9,10]. Transcriptomic profiling of lymphoblastic cells is a recent addition to the diagnostic toolbox and has led to the detection of new ALL subtypes [11,12,13,14,15]. Standard cytogenetic testing (G-banded karyotype) with adjunctive fluorescence in situ hybridization (FISH) can detect aneuploidy and chromosomal translocations such as *BCR::ABL1*, *ETV6::RUNX1* and *TCF3::PBX1* fusions [8,16]. *KMT2A* lesions, intrachromosomal amplification of chromosome 21 (iAMP21) and *IGH::CRLF2* and *P2RY8::CRLF2* fusions are also detectable. Other subtypes are identified only through additional analyses such as real-time PCR, single nucleotide polymorphisms and RNA sequencing. Subtypes such as *DUX4* rearrangements, *ETV6::RUNX1*-like and *PAX5* alterations are examples where the underlying genomic alterations are difficult to detect using standard-of-care laboratory methods, and RNA-seq profiling has emerged as an important diagnostic support tool [11,13,17]. For these reasons, understanding the information carried by transcriptomes in relation to the genome alterations in ALL is important, particularly for those patients for whom a conclusive genomic driver cannot be determined by current molecular diagnostics.

The high volume of RNA sequencing data per individual makes it necessary to employ machine learning techniques to process the raw information for the identification of subtypes [7,11,12,18,19,20]. In one such study, Gu et al. used clustering algorithms to characterize the transcriptional landscape of ALL [12]; the clusters were further investigated against identifiable genomic lesions to classify individual patients and refine the taxonomy of ALL. Using the taxonomy, the authors were able to assign subtypes for 94% of the study subjects. Recently, Schmidt et al. used the taxonomy to classify patients in multiple datasets [11,21]. Encouraged by these successes, we set out to leverage gene expression data within our transcriptomic analysis pipeline of B-cell ALL to improve diagnoses and gain biological insight into the transcriptional landscape of ALL.

Several issues related to mRNA-seq profiling of ALL have not yet been addressed. Firstly, clustering analyses of mRNA-seq data should be subjected to the rigorous adjustment of biases that is standard practice in epidemiology [22,23,24]; instead, most studies opt to (mis)use methods such as surrogate variable analysis that may cause artifacts if the data batches and biological characteristics are correlated (as they tend to be in most multi-cohort collections). These artifacts will be further amplified by clustering algorithms and machine learning models. Cohort biases may explain why gene signatures of subtypes derived from gene expression clusters, such as Ph-like, have been difficult to consolidate between different studies without additional experiments and analyses [25,26,27].

Secondly, a taxonomy that is based on gene expression profiles should not be used when fitting a machine learning model to RNA-seq read counts. Under the worst-case scenario, a transcriptome dataset affected by cohort bias leads to artificial clustering; the same artificial clustering of gene expression is then captured by machine learning and passed on as a falsely distinct subtype. Instead, we propose that gene expression classifiers should be trained with directly observable sequence variants or otherwise independently distinguishable subtypes or with longitudinal data on clinical outcomes and treatment effects.

The aim of this study is to introduce a reliable ALL classifier that can be integrated into current transcriptomic analyses with minimal additional resources and that can reliably classify ALL cases in an unbiased manner. To achieve reliability, we use new techniques to adjust for cohort and RNA-seq platform biases in a set of 1279 North American patients and then validate the predictions in an independent cohort of 767 Australian patients. To achieve easy integration into existing workflows, we introduce the Allspice R package with extensive documentation, small programmatic footprint and additional features for predicting genomic drivers and for confirming the tissue identity of the biological samples. The tool is also easy to train for any other disease or classification problem or to update with improved models of ALL in the future.

## 2. Results

### 2.1. Cohort Characteristics and Study Design

The distribution of ALL subtypes across cohorts is shown in Table 1. Note the definition of genomic subtypes: we excluded categories such as Ph-like due to the technical reasons described in Section 4 and in Appendix A. The most common subtypes included *CRLF2* (between 3.0% and 18.4% of cohort participants), *ETV6::RUNX1* (≥ 9.9% of pediatric patients), hyperdiploid (≥ 9.8% of pediatric patients), *KMT2A* (between 2.2% and 13.4%) and *BCR::ABL1* (between 2.4% and 21.5%). There were differences between the cohorts regarding age (*p* ≤ 7.6 × 10^−11^) and several subtypes, including pediatric *CRLF2* (P = 8.4 × 10^−12^), pediatric *ETV6::RUNX1* (*p* = 2.3 × 10^−8^), *KMT2A* across both age categories (*p* ≤ 3.2 × 10^−7^) and non-pediatric *BCR::ABL1* (*p* = 0.00016). The percentage of undefined samples was between 8.7% in the St. Jude Children’s Research Hospital cohort and 33.1% in the Australian pediatric cohort.

The RNA data were filtered and normalized separately for the North American and Australian samples (Figure 1A,B), and the training data were adjusted for technical and cohort confounders (Figure 1C). We also saved the unadjusted RNA data and compared it against the adjusted data to remove the most confounded genes (Figure 1D). In the next step, we created an internal training and testing set by splitting the North American data into two random subsets (Figure 1E). These two subsets were used as the initial material for selecting features and controlling for the complexity of machine learning models via hyperparameters. For the full models, we used all North American samples for training and the Australian dataset as an independent external validation set (Figure 1F).

### 2.2. Adjustments for Confounders

Confounding factors were mitigated first by surrogate variable analysis within pre-defined cohort batches (details in Section 4 and in Appendix A), and batch differences between the cohorts were then adjusted with a new approach we recently developed for time-series metabolomics data (accepted for publication). The adjustments removed correlations between gene expression and RNA library format (Figure 2A,D) and between the two continents of origin (Figure 2B,E) but did not influence the correlations between gene expression and ALL subtypes (Figure 2C,F; Appendix A).

To verify that the data processing was technically sound, we constructed a visual layout of the individuals according to the North American data (Figure 3A) and then projected the unadjusted American and Australian datasets onto the layout using the same statistical model (Figure 3B). The use of unadjusted data is important here since it provides a more realistic picture of how new, previously unseen samples would behave in a diagnostic pipeline. Clusters of the most frequent ALL subtypes were observable, and there was a high degree of concordance between the training and validation cohorts. A visualization of all subtypes and undefined samples is available in Appendix A. For convenience, we organized the subtypes into a central supergroup (*CRLF2*, hyperdiploid, hypodiploid, *PAX5* alterations and *BCR::ABL1*) and distinct subtypes on the periphery (*DUX4*, *ETV6::RUNX1*, *KMT2A*, *PAX5* P80R, *TCF3::PBX1* and *ZNF384*).

### 2.3. Classification of B-Cell ALL Subtypes Based on RNA-Seq Profiling

Positive predictive values (PPV) of machine learning models are visualized in Figure 4, and complete performance metrics are available in Appendix A. We used three different types of models (centroids, PLS and random forest, details in Methods) to exclude any artifacts that may be specific to a particular algorithm. Hyperparameters are listed in Appendix A. We focused on the external validation set as the primary benchmark of accuracy. Furthermore, all performance metrics were calculated for standardized but unadjusted data to simulate a scenario where new samples are analyzed one at a time without the opportunity to adjust for cohort effects.

Overall, differences between the three methods were negligible when considering the confidence intervals of the performance estimates (Figure 4). Accurate classification models were achieved for *DUX4* (PPV ≥ 95% in the external validation cohort across the three methods), *ETV6::RUNX1* (PPV ≥ 91%), *KMT2A* (PPV ≥ 84%), *PAX5* P80R (PPV ≥ 85%), *TCF3::PBX1* (PPV ≥ 92%) and *ZNF384* (PPV ≥ 96%). Together, 186 individuals (24%) of the Australian participants had one of these genomic subtypes (Figure 4D). More varied PPVs were observed in the central supergroup, including the *CRLF2* subtype (PPV ≥ 83%, Figure 4B), hyperdiploid (PPV ≥ 72%), hypodiploid (PPV ≥ 54%), *PAX5* alterations (PPV ≥ 33%) and *BCR::ABL1* (PPV ≥ 90%). Collectively, corresponding genomic subtypes were observed in 304 individuals or 40% of Australian participants (Figure 4D). Medium to high accuracy was achieved for rare subtypes such as *IKZF1* N159Y (PPV = 100%, Figure 4C); however, the small number of cases resulted in substantial statistical uncertainty.

### 2.4. Allspice Classifier

Based on the results, we concluded that the centroid model of 45 prioritized genes is the preferred choice as the practical classifier due to its comparable performance to the other methods and technical simplicity. The centroids were also robust against overfitting, as indicated by the flattening of internal training and testing performance when the number of inputs was increased (Appendix A). The robustness against overfitting enabled us to modify the study design to extract the maximum information from the available data (Appendix A). In the new design, the centroid model was fitted to the combined adjusted American and Australian data. We also added an extra step to account for sex and age that may carry important predictive information.

The results for a *BCR::ABL1* patient are depicted in Figure 5. The classifier identifies the subtype centroid that is the most similar to the observed RNA expression profile (Figure 5A). The display includes the frequency of the subtype in the training data versus any other subtype given the patient’s RNA profile (Figure 5B). Allspice also provides more detailed information on how the patient fits the transcriptional landscape of ALL (technical details in Section 4). The left panel shows the proximity of the patient to each subtype, respectively (Figure 5C). The *ETV6::RUNX1* subtype is an example of a genomic lesion that manifests as a clearly observable signature (Appendix A). On the other hand, there is more overlap within the central supergroup and subtypes such as hypodiploid can manifest simultaneous transcriptional proximity to multiple subtypes (Appendix A). If this overlap is too great, i.e., if it is uncertain statistically which subtype is the closest match, the patient is classified as having an ambiguous transcriptional profile (Appendix A).

The middle panel contains information on which combination of lesion-harboring genes best fits with the observed RNA profile (Figure 5D, see also Methods). In this case, the gene expression profile is compatible with the classical *BCR::ABL1* fusion without other strong signals. In the release version of Allspice, we have included only mutually exclusive gene combinations with at least five cases in the training set since rarer combinations could be difficult to confirm statistically. Matching affected genes directly with RNA profiles may provide additional clues for samples where other diagnostic results are inconclusive (Appendix A).

The right-hand panel shows the proximity of the RNA profile to typical B-cell ALL versus other cell types from public sources (Figure 5E). For example, the Australian cohorts include patients that were recruited from routine practice, some of whom had a low leukemia burden (Appendix A). For these individuals, the RNA data is closer to healthy blood and will be indicated by this panel. This feature is useful in circumstances where the sequence analyst has limited clinical information available.

### 2.5. Classifier Performance

The overall classification results are shown in Figure 6 and in Table 2. Of 2046 transcriptomes, 483 (24%) were designated ambiguous, and 89 (4.4%) were not classified due to poor proximity to any subtype. Performance was estimated first for all samples, including those with undefined genomic subtype. Since not all B-cell ALL cases can currently be attributed to a specific genomic lesion, these numbers are conservative estimates for the accuracy of the gene expression profile as an indicator of the underlying genomic lesion. Distinct subtypes were detected with high confidence (PPV ≥ 87%), whereas there were more ambiguous and unclassified samples in the central supergroup (see Appendix A for a detailed break-down).

The second set of results was calculated for 1598 patients that had a confirmed genomic subtype. Strong performance was observed for distinct subtypes (PPV ≥ 97%) and moderate accuracy for the central supergroup (PPV ≥ 75%). This scenario captures the statistical accuracy of the classifier in ideal conditions. Thirdly, when samples that failed the proximity or exclusivity thresholds were excluded, further improvements in PPV were seen across subtypes (10 out of 18 subtypes showed PPV = 100%). This shows how assessing the sample quality will help to avoid the misclassification of borderline cases. Of note, a high proportion of samples (56%) that were classified as having *BCL2/MYC* gene expression subtype were also identified as not originating from ALL B-cells in the tissue classifier (Appendix A), which may explain why it was the most difficult subtype to predict.

## 3. Discussion

B-cell ALL remains a life-threatening disease, particularly for adult patients of specific genomic subtypes [1,2,3,9]. Recently, rapid progress has been made in detecting ALL subtypes by RNA sequencing [12,13,14,15,19] and in subtype-specific treatments [6,20]. In this study, we present new data from a large Australian dataset and new findings from rigorous statistical and practical considerations to better leverage gene expression profiling in the diagnosis of ALL subtypes. We confirmed six genomically defined subtypes in one-third of patients that produce highly predictable mRNA profiles (PPV ≥ 89%). A further 40% of individuals were distinguishable by mRNA-seq expression levels, although the associations between specific genomic lesions and transcriptomic profiles were less certain. To dissect the biological ambiguity, we developed a proof-of-concept classifier that aggregates genome-wide mRNA-seq read counts into simplified RNA biomarker scores that indicate how well and where a patient’s RNA profile fits in the landscape of B-cell ALL subtypes.

### 3.1. Definition of ALL Subtypes

We used simpler definitions of genomic ALL subtypes compared to some of the previous reports [12,21]. The streamlined presentation provided multiple benefits, although trade-offs were unavoidable. Firstly, it allowed for sufficient group sizes for robust statistics and a statistically meaningful overview of the transcriptional landscape (e.g., Figure 3). On the other hand, more granular information on the exact nature of sequence alterations may be of high clinical importance but not captured by our subtype definitions. To gain better mechanistic insight, the Allspice classifier includes a feature that indicates genes that may harbor a driver mutation (Figure 5D), and further development of this concept may enable accurate diagnostics for targetable gene expression abnormalities. We also designed Allspice to support subtype definitions that are not mutually exclusive, thus providing flexibility for future updates.

The second rationale for the streamlined ALL taxonomy was to ensure a rigorous study design for machine learning. Previous studies have classified patients according to the way their RNA-seq expression profiles cluster (examples include the Ph-like and *ETV6::RUNX1*-like subtypes [12,13]). However, these definitions are problematic for the training of RNA profiling classifiers—the same gene expression levels should not be used to first define and then predict a subtype. Instead, we relied on observable sequence abnormalities or other information that was not derived from gene expression levels (except for *DUX4*). As a trade-off, the size of the training set decreased, and the number of individuals with undefined genomic subtypes increased; however, such uncertainty is to be expected in real-world datasets that manifest substantial biological variation in how genomic lesions drive altered gene expression profiles.

### 3.2. Classification Performance and Utility

Overall, B-cell ALL subtypes that could be identified by fusion callers and cytogenetics had distinctive mRNA-seq read count profiles (Allspice classified 90% of samples with a defined genomic subtype correctly). In a recent study that used mostly the same datasets, the correct classification rate was between 82% and 93% [21] and similar rates have been reported in other machine learning studies of ALL [11,19,28,29,30]. Therefore, the performance of the Allspice tool is within the range of other similar classifiers, which demonstrates the rich biological information available from RNA-seq data and the stability of the predictions across multiple types and implementations of classifiers.

If a patient tests positive by Allspice for one of the six distinct subtypes (*DUX4*, *ETV6::RUNX1*, *KMT2A*, *PAX5* P80R, *TCF3::PBX1* and *ZNF384*), our findings suggest that the subtype can be validated by deeper exploration of the sequencing reads in the same RNA-seq dataset or by molecular diagnostics for at least 89% and up to 99% of cases depending on the subtype. Both the sequencing and molecular analyses can be time-consuming and inconclusive, whereas mRNA expression levels (i.e., inputs to Allspice) can be reproducibly calculated using highly standardized algorithms. This will shorten the time to diagnosis for the vast majority of ALL cases with the aforementioned recurrent lesions, and significantly shorten the time to delivering care in the clinical setting. Identification of other lesions such as *CRLF2*, hyperdiploid, hypodiploid, *PAX5* alterations and *BCR::ABL1* are less definitive, though this may improve with further algorithm training and refinement. There may be clinical utility for the Allspice biomarker panels as RNA risk factors for adverse outcomes in patients that show abnormal karyotypes.

A total of 448 patients lacked a definitive genomic subtype under the streamlined taxonomy used in this study. Diagnostics for this patient subpopulation is where we expect mRNA-seq profiling to provide the best added value. In this respect, Allspice is a unique tool since it also provides quantitative RNA biomarkers for the most likely driver lesions and for the deviation from the healthy blood transcriptome. These features are particularly useful when the ALL subtype is difficult to establish due to the absence of identifiable sequence alterations or inconclusive cytogenetic findings. Based on the RNA data, we assigned a transcriptionally compatible ALL subtype to 207 patients (41% of 448) —these are conceptually similar to the Ph-like and *ETV6::RUNX1*-like gene expression profiles from previous studies. The same concept can be extended to genomic drivers. For example, there were 90 (20%) patients with simultaneous alterations in CRLF2 and P2RY8 and 79 (18%) patients with alterations in both IGH and BCL2 among the undefined subpopulation. Given the sizes of these secondary subgroups, they may be considered as additions or replacements for historical subtypes as genomic ALL datasets to get larger and better phenotyped. The gene expression profiles of 57 (13%) patients could not be matched with any typical ALL subtype, and we suspect many of these were individuals with a low leukemia burden.

### 3.3. Strengths and Weaknesses

The large sample size (statistical power), diversity of data sources, careful mitigation of potential confounding factors and comparisons between three classes of machine learning algorithms make this study strong from a methodological perspective. Notably, the Australian samples were collected from routine health care settings, which provides a realistic spread of sample quality and leukemia burden as encountered in clinical practice. Furthermore, we used additional datasets to help assess the quality of the samples and safety against misclassification, which is an important practical consideration outside research settings [31]. The data were obtained from three Western countries, and caution is warranted if applying the findings in a different ethnic or socioeconomic context. We included rare subtypes in Allspice; however, these predictions should be interpreted with caution since we could not determine classification performance accurately (Figure 4C). Furthermore, we excluded genes that were expressed only in a few samples, some of which may have been highly indicative of a rare subtype. Currently, the driver gene feature of Allspice is based on limited information about the most important lesions observed in an individual, and the results should be interpreted as suggestive rather than definitive regarding causality. Due to the careful analyses and robust performance, we anticipate that the classifier we created captured biological information that reflects the causal mechanisms of ALL and is, therefore, likely to work well for most patient populations.

### 3.4. Practical Considerations

Allspice is open source, easy to install on the popular R programming environment via the Comprehensive R Archive Network and comes with extensive documentation. It accepts raw read count data as produced by standard RNA sequencing pipelines, which is an advantage in clinical settings that may lack a dedicated bioinformatician. The R environment already includes tools for visual clustering of transcriptomes using algorithms such as t-SNE [32], but clustering results can be difficult to interpret for individual cases. Rather than relying on visual proximity in a scatter plot, Allspice uses quantitative probabilistic metrics to indicate the certainty of the predicted subtype. Furthermore, the ability to analyze one sample at a time is important: t-SNE or hierarchical cluster analysis are designed for the research space where large cohorts of labeled samples are readily available, whereas Allspice was designed for a single sample from the beginning.

We included two examples with unusual lesions where Allspice helped to assign a (transcriptional) subtype. The first case was an individual with an undefined genomic subtype that Allspice classified as having an RNA profile compatible with an ETV6::RUNX1 fusion. Uncommon ETV6 fusions were discovered by detailed investigations (Appendix A). Another individual was classified as having a ZNF384-like transcriptional profile, while the exact causal lesion remained uncertain (Appendix A). These examples highlight how the additional information from Allspice can guide diagnostic efforts for patients with unusual genomic lesions.

## 4. Materials and Methods

### 4.1. North American Participants

A total of 649 males, 541 females and 89 patients without gender information were from St Jude Children’s Research Hospital (St Jude); Children’s Oncology Group (COG); ECOG-AGRIN Cancer Research Group (ECOG-AGRIN); MD Anderson Cancer Center (MDACC); the Alliance of Clinical Trials in Oncology, Cancer and Leukemia Group B (CALGB) and University of Toronto (Toronto). Detailed clinical information for each case and listings of clinical trial numbers have been previously published [12]. RNA-seq data files were obtained from the European Genome-Phenome Archive (EGAD00001004461 and EGAD00001004463). The patients who participated in this study have provided written informed consent, assent (as appropriate) or parental consent (as appropriate) as part of enrolment protocols for research, including genetic research. All relevant ethical regulations were followed during this study.

### 4.2. Australian Participants

A total of 387 males, 278 females and 102 patients without gender information were investigated through the Australasian Leukaemia and Lymphoma Group (ALLG) National Blood Cancer Registry and the associated Regalia project (ACTRN12612000337875), Australian & New Zealand Children’s Haematology/Oncology Group Acute Lymphoblastic Leukaemia Study 8 (ACTRN12607000302459) and Study 9 (ACTRN12611001233910). All protocols had been approved by relevant human research ethics committees.

### 4.3. Supporting RNA Data

RNA data were sourced from a previously published study of 660 lymphoblast cell lines [33] and from the Genotype-Tissue Expression (GTEx) project release 8 [34]. The GTEx includes RNA-seq data from 948 donors and 54 tissues. In this study, we organized the data into 31 organ groups, of which those that contained at least 500 samples were selected, including whole blood as the most relevant tissue type for ALL.

### 4.4. RNA Sequencing and Pre-Processing

RNA analyses of the North American samples have been described previously [12]. Briefly, RNA-seq was performed using TruSeq library preparation and HiSeq 2000 and 2500 sequencers (Illumina Inc., San Diego, CA, USA). All sequence reads were paired-end and were obtained from total RNA and stranded RNA-seq (75 or 100 base-pair reads) and polyA-selected mRNA (50, 75 or 100 base-pair reads). In Australia, library preparation for mRNA sequencing was performed using either Truseq Stranded mRNA LT Kit (Illumina Inc., San Diego, CA, USA) or Universal Plus mRNA-Seq with NuQuant (Tecan Group Ltd., Männedorf, Switzerland) from total RNA as per the manufacturer’s instructions. Samples were sequenced by either HiSeq 2000 or NextSeq 500 platforms (Illumina Inc., San Diego, CA, USA) producing 75b length paired-end (PE) reads with a median read depth of 65M reads.

Raw reads from all cohorts were aligned and mapped to the GRCh37 reference genome with the STAR software version 2.4.2a and above using the two-pass mode [35]. Raw gene counts were generated from BAM files using featureCounts [36]. We defined a gene to be usable as a potential biomarker if it had a read count of ≥ 100 in at least 1% of samples in both the North American and Australian datasets, respectively. In total, 18,923 genes were included in the study. Expression counts were normalized using the DESeq2 algorithm [37], and the normalized counts were transformed using the formula *log_2_(count + 1)* before statistical analyses.

### 4.5. Genomic Subtyping

In the text, we use the term ‘genomic ALL subtype’ when the subtype was assigned according to a DNA or RNA sequence abnormality or a clinical biomarker independently of gene expression levels (except for DUX4, PAX5 alterations and CDX2). The detection of genomic alterations and subsequent subtyping of ALL cases were based on the previously published analyses of the North American samples [12] and a preliminary definition of a recently discovered rare CDX2 subtype [38]. We were not able to define the DUX4 subtype independently of mRNA expression levels due to its cryptic nature. PAX5 alterations were defined partly by gene expression data (North America) or by directly observable PAX5 alterations for samples that could not be otherwise classified (Australia).

Genomic alterations in the Australian samples were detected as follows. FusionCatcher [39], SOAPfuse [40] and JAFFA [41] were utilized to identify clinically relevant gene fusions. Only fusions reported by multiple fusion calling algorithms were considered, with the exception of rearrangements involving the IGH locus (IGH-DUX4), which were confirmed by high levels of DUX4 expression [42] or by manual inspection of the fusion and accompanying expression data. Single nucleotide variants were identified with GATK-haplotype caller [43] following the best practices workflow. Copy number alterations were detected by multiplex ligation-dependent probe amplification using two SALSA Reference kits (P335 and P202, MRC-Holland, Amsterdam, Netherlands), according to the manufacturer’s instructions.

To harmonize subtype definitions between the cohorts and to include only the most confident classifications, we relabeled subtypes for the statistical analyses (Appendix A, Appendix A).

In addition to the pre-defined subtypes, we determined every gene and combinations of genes that harbored an alteration for any individual. The combinations were sourced computationally from the metadata by first collecting all the gene symbols included in the reported rearrangements or mutations. The labels were then collected as a list, and unique combinations were labeled as pseudo-subtypes. For example, an individual with the gene fusion *BCR::ABL1* without additional alterations received the label ‘ABL1,BCR’ to indicate the two affected genes. However, an individual with KMT2A::MLLT1 and ID2::IGK fusions would receive the label ‘KMT2A,MLLT1,ID2,IGK’; thus, the gene combinations should not be automatically interpreted as fusions. These mutational profiles were also used for creating genetically matched subsets to allow for more accurate batch adjustments (details below).

### 4.6. Adjustments for Confounders

We divided the data into five batches according to country of origin and age of the patients (Table 1). We then used statistical adjustments to mitigate the potentially confounding associations between data sources and ALL subtypes. Firstly, we used Surrogate Variable Analysis [44] to reduce undesired variation in normalized read counts within each batch, respectively. Next, we used genetically matched subsets and the Numero R package [45] to remove potentially confounding variations between the batches (Appendix A). Genes that were perfectly aligned with a batch were excluded, which left 18,503 adjusted genes (98%) for statistical analyses. We then calculated correlations between unadjusted and adjusted versions of gene expression and excluded unstable genes that showed a Pearson correlation of R < 0.9 (Appendix A). A total of 6673 genes were considered stable and included as inputs to classification models.

### 4.7. Machine Learning

We chose a random forest approach [46] as an example of a supervised non-linear machine learning technique that can predict multi-class outcomes from complex input data. We also created Projections to Latent Structures (PLS) for each ALL subtype separately as an example of a linear factorization method [47]. Separate PLS models were fitted to each subtype versus other samples. The third type of modeling was based on neighbor distances: we calculated mean RNA profiles (centroids) for every ALL subtype and classified individuals based on the nearest centroid in the data space. Unsupervised clustering was achieved with the Uniform Manifold Approximation and Projection (UMAP) algorithm [48] to gain qualitative visual insight into the RNA-based segregation of ALL subtypes.

Input data were standardized the same way for each model using the default settings of the function ‘numero.prepare()’ in the Numero R package [45]. This is a refined version of the empirical Z-score with protections in place for outliers and skewed distributions. Normalization and standardization parameters were determined for the training dataset and applied independently to the external validation set to simulate a scenario where new unseen data are analyzed one sample at a time and thus must be pre-processed using pre-defined parameters.

### 4.8. Pruning of Correlated Input Features

To improve the performance of UMAP and nearest centroids, the full list of detectable genes was pruned using an approach similar to clumping in genetics [49] and the pruned set of genes was used as input features. First, we calculated Welch’s t-statistics for each gene and ALL subtype and converted them to Z-scores using the cumulative t-distribution and inverse cumulative Normal distribution. Next, we calculated the variance of the Z-scores for each gene as an aggregate measure of how well a gene segregated between ALL subtypes. Genes were then sorted from large to small variance. In the final step, the sorted list was traversed while checking if the next gene was correlated (R ≤ −0.3 or R ≥ 0.3) with any of the already selected. The Australian data were excluded from the pruning procedure to ensure independent external validation.

### 4.9. Training, Testing, External Validation and Performance Metrics

The division of data for the evaluation of classification performance is shown in Figure 1. To determine the optimal model complexity (hyperparameters), North American participants were divided into two randomized subsets that were used as internal training and testing sets. The randomization was completed separately for each subtype to ensure matching subtype frequencies. Next, models were fitted to one of the subsets (internal training set) and classification performance was evaluated in the other (internal testing set). Classification performance in the testing set was used to determine optimal hyperparameter settings (Figure 1E).

Full models were trained with the full North American dataset and validated externally in the Australian dataset. We focused on positive predictive value (PPV) as the primary performance metric due to its suitability for the low case frequencies of most ALL subtypes. Negative predictive values, sensitivity, specificity and the area under the receiver operating characteristic curve were also calculated.

### 4.10. Proximity and Exclusivity

The open-source classification tool Allspice is available in the Comprehensive R Archive network (URL: https://cran.r-project.org, 18 April 2022). It includes a visual display of the classification results (Appendix A) and two quality measures to help decide if an RNA profile indicates a specific ALL subtype. The proximity measure is the output value from the probit regression step in the Allspice modeling design (Appendix A) and represents the likelihood of observing the subtype in the training population when balanced for group sizes and given the observed RNA biomarker value and covariates. We chose 50% as the threshold for acceptable proximity. To identify samples with mixed subtype characteristics (designated as ‘Ambiguous’), we defined exclusivity as the difference in the proximity scores between the best and second-best matching subtype centroids. The χ^2^-distribution with one degree of freedom was used to convert the difference into a probability. We chose 50% as the default threshold for acceptable exclusivity.

## 5. Conclusions

We observed strong associations between genomic alterations and lymphoblast transcriptome-wide expression profiles in pediatric and adult patients of B-cell ALL. For one-third of patients, these associations are unambiguous and provide diagnostic information that is often quicker and easier to obtain compared to fusion callers or cytogenetic tests. For the rest of the patients, gene expression analyses may provide insight that is not available from other methods. An RNA-based ALL biomarker can inform sequence analysts where to look for lesions manually if automatic fusion callers failed or read counts were too low for statistical certainty. In both scenarios, Allspice can help oncologists determine the most likely causal drivers with greater confidence and identify potential therapeutic targets in a shorter time frame.

## Figures and Tables

**Figure 1 ijms-23-04574-f001:**
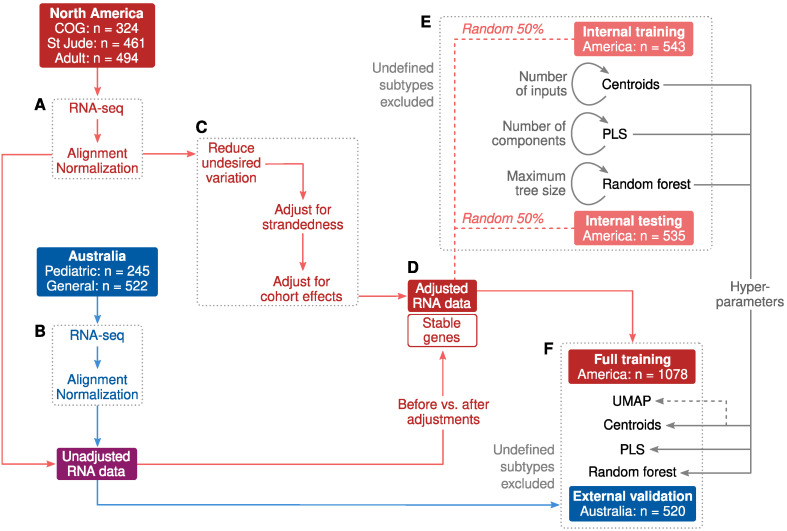
Study design. (**A**,**B**) RNA-seq pre-processing was applied separately for North American and Australian datasets. (**C**) Non-biological differences due to technical artifacts and cohort effects were adjusted according to genetically matched subsets (details in Methods). (**D**) Correlation coefficients were calculated between unadjusted and adjusted expression levels to exclude genes that were heavily influenced by confounders. A total of 6673 stable genes with R ≥ 0.9 were included for subtype modeling. (**E**) Internal training and testing sets were randomly chosen from the North American participants as initial material to evaluate machine learning models and to determine hyperparameters. The randomization was conducted separately for each subtype to ensure matching subtype frequencies, which explains the small difference in set sizes. (**F**) Final machine learning models were trained with the full North American dataset and validated in the Australian dataset.

**Figure 2 ijms-23-04574-f002:**
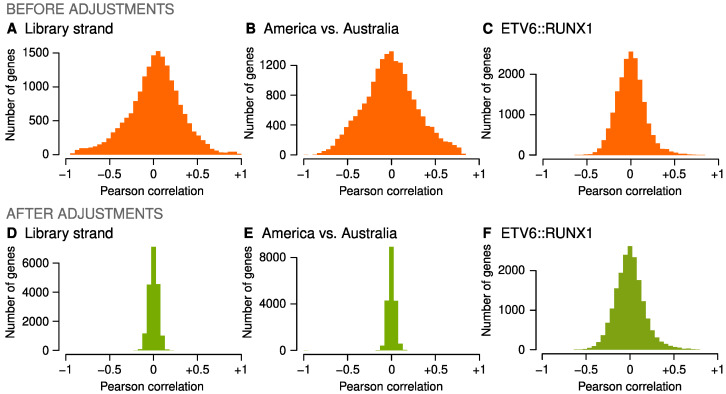
Impact of confounder adjustments. Pearson correlation coefficients were calculated between 18,503 log-transformed genes, and a technical or clinical variable before and after gene expression levels were adjusted as described in Methods. A wide histogram indicates substantial covariation across the transcriptome, while a narrow histogram indicates successful removal of covariance. (**A**,**D**) The North American cohorts included 204 (16%) RNA samples that were sequenced with an unstranded library. (**B**,**E**) Strandedness and other technical differences manifested as substantial covariation between the transcriptome and the continent of origin. (**C**,**F**) Covariation between gene expression and ALL subtypes, such as ETV6::RUNX1, was preserved.

**Figure 3 ijms-23-04574-f003:**
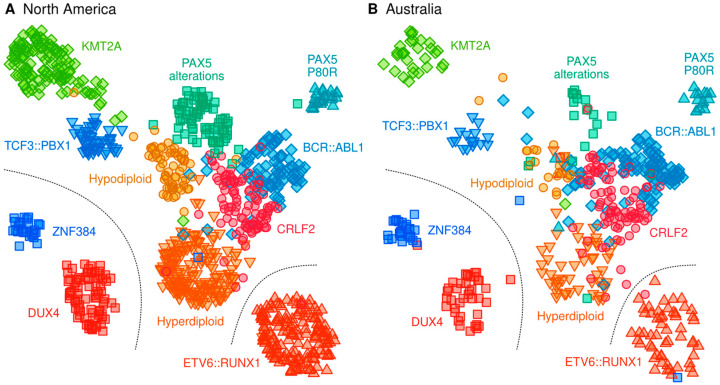
Transcriptional landscape of the most frequent ALL subtypes. Clustering structure in the North American data was modeled by the Uniform Manifold Approximation and Projection (UMAP) algorithm and using the same genes that were prioritized and pre-processed for the centroid classifier. (**A**) Standardized but unadjusted gene expression values were used for re-projecting the North American samples onto the UMAP layout. We use the unadjusted expression profiles here since, in practical settings where patients arrive one-by-one, adjustments for batch effects that would be available in research settings cannot be made. (**B**) Unadjusted Australian data were projected onto the same UMAP layout as an independent external validation set.

**Figure 4 ijms-23-04574-f004:**
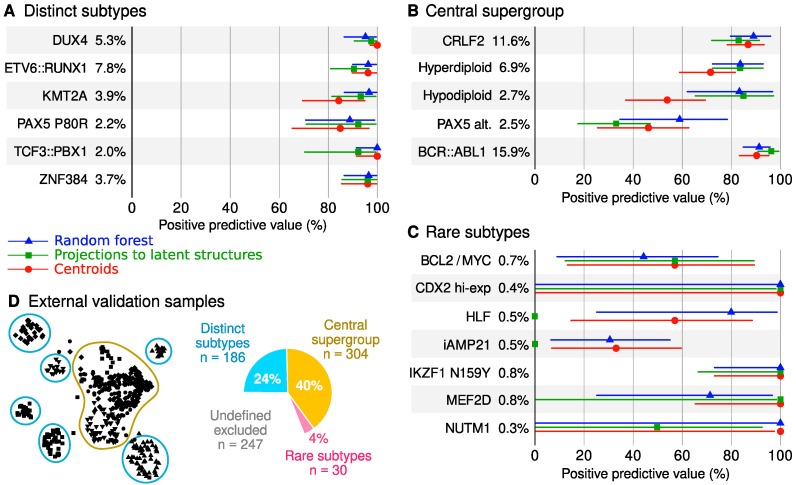
Comparison of three machine learning algorithms. Each model was fit to the batch-corrected North American data (training set *n* = 1078) and then evaluated in unadjusted Australian data (external validation set *n* = 520). Samples with undefined genetic subtypes were excluded from the analyses. (**A**–**C**) The forest plots show 95% confidence intervals that reflect the statistical uncertainty due to finite category sizes. The percentages written in the plots indicate the prevalence of the genetic subtype in the Australian dataset. (**D**) Description of subtype supergroups.

**Figure 5 ijms-23-04574-f005:**
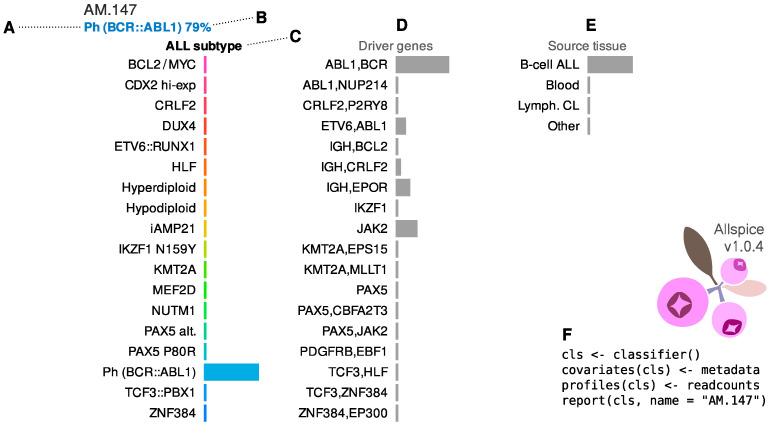
Example of a report card from the Allspice classifier for an adult male patient from North America with Philadelphia (*BCR::ABL1*) genetic B-cell ALL subtype. (**A**) Sample identifier and predicted subtype based on RNA data in top-left corner. (**B**) The report shows the frequency of the corresponding genetic subtype in the training data, given the observed gene expression profile. In this case, there was a 79% chance that sequence and cytogenetics analyses would confirm the presence of the Philadelphia chromosome. (**C**) Visualization of how similar the sample is to each ALL subtype profile. The display can be interpreted as a panel of RNA “biomarkers” that are specific to each subtype. In this case, the high value for Ph indicates that the gene expression profile is compatible with a typical patient with the Philadelphia chromosome. (**D**) Allspice also indicates how similar the sample is to the RNA profiles associated with the presence of genetic alterations in one or more genes in parallel. In this case, the gene expression profile matches the typical profile of patients with independently verified *BCR::ABL1* fusion (i.e., both *BCR* and *ABL1* are altered). (**E**) Samples with high leukemia burden will typically produce a strong B-cell ALL signal in the tissue panel. (**F**) Minimal example of how to generate the report in the R programming environment.

**Figure 6 ijms-23-04574-f006:**
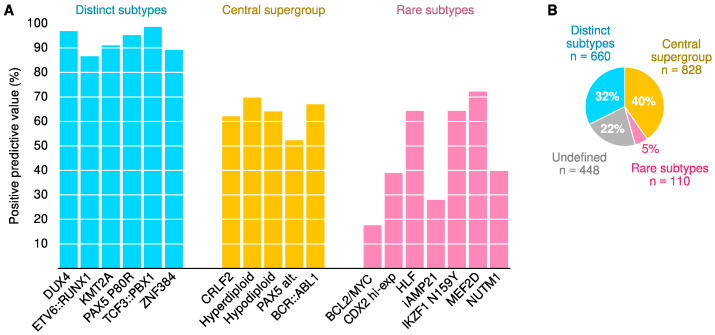
Classification performance of the Allspice centroid classifier. (**A**) The bars show positive predictive values for all samples including those with undefined genetic subtypes. They represent conservative estimates on how likely it is that the subtype predicted by RNA-seq expression levels can be confirmed as a specific sequence alteration or is also indicated by cytogenetics. (**B**) Proportions of genetic subtypes in the dataset.

**Table 1 ijms-23-04574-t001:** Patient characteristics and frequencies (%) of ALL subtypes according to known genetic alterations. The participants were grouped primarily by the recruiting institute and secondarily by age (> 99% of participants in the pediatric cohorts were below 20 years of age). Mean and standard deviation are shown for age. *p*-values for cohort differences were calculated by the χ^2^-test.

	Pediatric Cohorts				Adult and General Cohorts		
	COG	St Jude	Australia	*p*-Value	Multiple *	Australia	*p*-Value
Male	177	221	83	4.4 × 10^−6^	251	304	0.021
Female	120	188	73	0.016	233	205	0.013
Unknown	27	52	89	1.8 × 10^−21^	10	13	0.77
Age (years)	9.2 ± 5.7	6.4 ± 4.4	7.4 ± 4.0	7.6 × 10^−11^	45.1 ± 14.8	35.7 ± 22.6	3.6 × 10^−13^
BCL2/MYC	0.0	0.4	0.4	0.50	2.8	0.8	0.024
CDX2 hi-exp	0.3	0.0	0.0	0.34	0.6	0.6	1.0
CRLF2	7.4	3.0	18.4	8.4 × 10^−12^	13.2	8.4	0.020
DUX4	7.4	7.4	2.0	0.0096	4.9	6.9	0.21
ETV6::RUNX1	9.9	26.7	17.6	2.3 × 10^−8^	1.0	2.3	0.18
HLF	0.0	0.4	0.4	0.50	0.6	0.6	1.0
Hyperdiploid	21.0	21.3	9.8	0.00034	4.0	5.6	0.27
Hypodiploid	1.2	0.7	0.8	0.68	11.5	3.6	3.1 × 10^−6^
iAMP21	6.5	0.4	0.8	0.72 × 10^−8^	0.2	0.4	1.0
IKZF1 N159Y	0.3	0.0	1.2	0.043	0.4	0.6	1.0
KMT2A	2.2	10.8	2.9	2.0 × 10^−7^	13.8	4.4	3.2 × 10^−7^
MEF2D	2.2	1.1	0.0	0.058	2.0	1.1	0.39
NUTM1	0.3	1.1	0.4	0.36	0.0	0.2	1.0
PAX5 Alt	4.0	5.9	3.7	0.32	8.3	1.9	6.4 × 10^−6^
PAX5 P80R	0.9	1.1	0.8	0.94	3.4	2.9	0.74
BCR::ABL1	5.6	2.4	4.1	0.070	12.3	21.5	1.6 × 10^−4^
TCF3::PBX1	2.2	6.7	1.2	0.00023	3.0	2.3	0.59
ZNF384	1.9	2.0	2.0	0.99	2.8	4.4	0.24
Undefined	26.9	8.7	33.5	1.7 × 10^−16^	15.0	31.6	7.0 × 10^−9^

* ECOG-ACRIN, Toronto, MDACC and CALGB.

**Table 2 ijms-23-04574-t002:** Positive predictive values for correct classification into genetically defined B-cell ALL subtypes. The values are presented as the percentages of samples for which the best matching RNA centroid was the same as the genetic subtype if defined. Quality control was set at ≥ 50% proximity and ≥ 50% exclusivity (details in Methods).

	All Samples	Defined Subtypes	Defined Subtypes and QC Pass
Number of samples	2046	1598	1292
DUX4 (%)	96.8	100.0	100.0
ETV6::RUNX1 (%)	86.6	97.7	98.6
KMT2A (%)	91.1	96.6	100.0
PAX5 P80R (%)	95.2	97.6	100.0
TCF3::PBX1 (%)	98.5	98.5	100.0
ZNF384 (%)	89.1	96.6	100.0
CRLF2 (%)	62.0	91.2	97.6
Hyperdiploid (%)	69.8	92.1	99.0
Hypodiploid (%)	64.2	79.0	89.2
PAX5 Alt (%)	52.1	74.8	92.7
BCR::ABL1 (%)	67.2	84.8	96.6
BCL2/MYC (%)	17.5	54.1	66.7
CDX2 hi-exp (%)	38.9	77.8	100.0
HLF (%)	64.3	75.0	90.0
iAMP21 (%)	27.9	66.7	100.0
IKZF1 N159Y (%)	64.3	90.0	100.0
MEF2D (%)	72.2	78.8	100.0
NUTM1 (%)	40.0	66.7	100.0

## Data Availability

The Allspice software and source code are available via the Comprehensive R Archive Network (URL: https://cran.r-project.org, 18 April 2022). The North American mRNA-seq data files are available in the European Genome-Phenome Archive (EGAD00001004461 and EGAD00001004463). The Australian dataset is available from the authors upon request.

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
