# Peer review of "Multi-Cohort Transcriptomic Subtyping of B-Cell Acute Lymphoblastic Leukemia"

_ijms, 2022, doi:10.3390/ijms23094574_

Round 1

Reviewer 1 Report

In the manuscript ijms-1654485 “Multi-cohort transcriptomic subtyping of B-cell acute lymphoblastic leukemia” Mäkinen et al tested three different machine learning algorithms to establish a classifier for B-cell acute lymphoblastic leukemia (ALL) into major subtypes according to their mRNA-profiles. The resulting classifier is available as open source package (Allspice) in R.

It has been shown that B-ALL can be classified into more than 20 subtypes which might have clinical implications. Some of them are driven by gene rearrangements, aneuploidy or point mutations, but there are also subtypes that cannot be detected via routine diagnostics. RNA-seq is therefore more and more commonly used to further classify B-ALL. However, often bigger cohorts and trained bioinformaticians are needed to facilitate this work. As those requirements are not always met, in my opinion, the classifier developed in this study provides an easy-to-use tool where one can use single samples to at least give a first estimate of the subtype of unknown samples. With some basic bioinformatics knowledge the R-package can be installed, and the raw count data can be used to run the classification.

In contrast to e.g. AllSorts the classifier not only supplies information about the ALL subtype itself, it also gives an estimate of the potential driver genes, which might be very helpful to further explore the RNA-seq data, fusion detection tool results or perform validation in the lab.

The technical part does not meet my expertise, I therefore can not judge the correctness of the use of the machine learning algorithms or the adjustment for confounding factors used.

Comments

The authors claim that they did not use Ph-like or ER-like for classification as these subtypes are defined using the RNA-seq data. However, the same is at least partially the case for the PAX5-alt subtype. The authors should comment on how they define PAX5-alt. Did they specifically select samples that e.g. carry a PAX5-fusion genes or deletions or mutations?

A cutoff of 100 reads in at least 1% of the samples has been used (line 441) to include a gene as potential biomarker for one subtype. However, there are several subtypes comprising less than 1% of samples in the study therefore potential marker genes for small subgroups might be excluded prior to classification.

Please use the up-to-date nomenclature on fusion genes see Bruford et al, Leukemia, 2021.

The authors should comment on how they came up with the potential driver genes. What types of alterations are included? What does IKZF1, JAK2, PAX5 mean? Does this mean the gene is mutated, deleted or overexpressed? Furthermore, please check the names of the driver genes e.g. BCR::ABL1 and not ABL1::BCR is the main driver in Ph+ leukemia.

Table1: if calculating e.g. 0.4 % of the total number of Male + Female + Unknown does not yield an integer. Furthermore, the sum of subtype percentages is not 100%. Please verify the numbers.

Line 434: “1 g of total RNA” I assume less total RNA was used.

Reviewer 2 Report

Mäkinen et al. present a new and interesting tool to classify ALL and improve diagnosis. Information about B-ALL variation based on the fusion mutations are quite important. The authors showed lots of results regarding the fusion mutations. This comprehensive analyses are informative for the IJMS readers.

Author Response

REVIEWER 2: “Mäkinen et al. present a new and interesting tool to classify ALL and improve diagnosis. Information about B-ALL variation based on the fusion mutations are quite important. The authors showed lots of results regarding the fusion mutations. This comprehensive analyses are informative for the IJMS readers.”

AUTHOR response: Thank you for these positive comments! See also response to Comment 4 by Reviewer 1 on the description of driver genes.

Reviewer 3 Report

This paper includes original informative data about leukemia.The sample number is large and it is the precious real-world data.

Author Response

REVIEWER 3: “This paper includes original informative data about leukemia.The sample number is large and it is the precious real-world data.”

AUTHOR response: Thank you for the supportive statement. We made extra effort to ensure the results are applicable for real-world samples.